# Influence of Cold Environments on Growth, Antioxidant Status, Immunity and Expression of Related Genes in Lambs

**DOI:** 10.3390/ani12192535

**Published:** 2022-09-22

**Authors:** Lulu Shi, Yuanqing Xu, Xiao Jin, Zheqi Wang, Chenyu Mao, Shiwei Guo, Sumei Yan, Binlin Shi

**Affiliations:** College of Animal Science, Inner Mongolia Agricultural University, Hohhot 010018, China

**Keywords:** cold environment, growth performance, immune function, antioxidant status, gene expression, lamb

## Abstract

**Simple Summary:**

In the scenario of global climatic change, extreme weather is a severe threat to organisms worldwide. For the purposes of dissecting the potential basis of cold environment stress responses and adaptation in lambs in cold climates, we had used cold environments with different temperature gradients to assess the effect of cold environments on growth, antioxidant status and immune function in lambs. The experimental results showed that the growth, antioxidant status and immune capacity of lambs were remarkably inhibited cold environments.

**Abstract:**

Cold climates may be a risk to the health and welfare of lambs during winter because cold environments alter the physiological processes of lambs, and we used cold environments with three different temperature gradients—an indoor heating control group (IHC) using electric heating; an indoor temperature group (IT) with intermittent and slight degrees of stimulation of coldness; an outdoor temperature group (OT) exposed to cold environments in an external natural environment. The results showed that the lambs in the OT group had a greater decrease in the average daily gain (ADG) and increase in the average daily feed intake (ADFI) and the feed-to-gain ratio (F:G) compared to the other two groups. The decrease in immunoglobulin A (IgA) and interleukin 2 (IL-2) contents and *IL-2* gene expression, and the increase in tumor necrosis factor α (TNF-α) content and *TNF-α* and nuclear factor kappa-B p65 (*NF-κB p65*) gene expressions in the OT group suggested that the lambs had a compromised immune status in cold environments. Moreover, the decrease in catalase (CAT), glutathione peroxidase (GPx), total superoxide dismutase (T-SOD), and total antioxidant capacity (T-AOC) levels, and *CAT*, *GPx*, *SOD1*, *SOD2*, and nuclear factor-erythroid 2-related factor 2 (*Nrf2*) gene expressions, and the increase in malondialdehyde (MDA) in the OT group suggested that the lambs had a lower antioxidant defense capacity in cold environments. Thus, in extreme cold, lambs kept outdoors could reduce growth, immune function and antioxidant status. However, shelter feeding in winter could relieve the stress of cold environments on lambs, and housing with heating equipment was more conducive to the improve growth, immune, and antioxidant function of the lambs.

## 1. Introduction

As an important external factor to constrain the development of livestock production, the ambient temperature can directly influence the physiological processes of animals or indirectly alter the quantity and quality of feeds available to them. The range of environmental temperature at which organisms can maintain a constant body temperature with the least involvement of thermoregulation is called the thermo-neutral zone [1], while an environmental temperature beyond the upper and lower limit of the zone of thermo-neutrality is supposed to generate heat or cold stress in animals [2]. One of the major potential stressors of livestock in winter, cold stress induces physiological responses which are of high priority and energy demand for homeotherms. Under cold exposure, homoeothermic animals need to maintain a constant body temperature by generating additional heat from their diet and body to cope with cold stress, which increases the energy demand of the animals [3]. Non-ruminants have different carbohydrate metabolisms, in comparison to ruminants, whose dietary glucose is converted into short-chain volatile fatty acids in the rumen, so the main source of blood glucose is gluconeogenesis in the liver, even in the fetal liver [4]. This may result in blood glucose concentrations lower than that in non-ruminant animals, and thus predispose the animal to a negative energy balance when energy demand rises. This condition can progress into various disorders of energy metabolism and possibly disease in ruminants [5]. Alternatively, cold exposure is characterized by an elevated metabolic heat production rate and this possibly also disrupts the balance of the oxidant/antioxidant system as a consequence of the increased production of reactive oxygen species (ROS), causing lipid peroxidation and oxidative stress [6]. It is well recognized that oxidative stress can lead to oxidation of biomacromolecules and disruption of normal metabolism and physiology, which is crucial to the development and exacerbation of the symptoms of numerous diseases [7,8]. Considering these facts, exposure to extreme temperatures, whether a cold or heat environment, causes additional discomfort and increases stress levels and thus can adversely compromise the health, reproduction and productivity of domestic animals, especially ruminants [9].

Sheep, due to their multifaceted utility for milk, meat, wool, skin, and manure, are an important livestock species contributing greatly to food, fiber, rural employment, and financial income in arid and semi-arid regions [10]. The worldwide importance of sheep is well established due to their advantageous traits, including the ability to thrive in extreme harsh environments, drought and famine [11]. Sheep therefore play an important role in the global agrarian economy, especially in the changing climatic scenario. Sheep are considered to have a wide range of ecological adaptability due to specific anatomical and physiological characteristics, which lends them higher productivity in hard environmental conditions than other livestock [12]. However, this might not remain true in the context of very cold conditions. Some animal owners believe that heavy cold winters can be stressful and even fatal for sheep, particularly lambs, which show hypothermia, and result in hypoxia, hypoglycemia, metabolic acidosis and alterations in the metabolism of water and electrolytes [13]. The large temperature difference between day and night during the winter and a long-term cold environment may cause sheep to experience extreme cold stress. In such circumstances, it is quite possible to observe a depression of yield in sheep as a result of the substantial dietary energy intake being diverted from productive functions to thermoregulation. More severely, cold stress can cause the development of secondary changes and possibly disease or even death when the organism fails to produce sufficient heat [14]. While the critical temperature of endothermy may vary for different breeds, productive orientation, level of health, and the nature, intensity, frequency and duration of the stressor, Curtis [15] recommended values of 5 and 25 °C as lower and upper limits for the thermo-neutral zone for ovine species in normal circumstances. A previous study proposed that the lower critical temperature of adult sheep, with reasonable wool coverage (2 cm), is −3 °C [16]. In addition, for adult ewes in pastoral regions of northwest China, a temperature below 2 °C during the cold season is not advised [17]. In actuality, however, for the most part, sheep are raised on uninsulated and unheated brans and therefore at temperatures far below this in the winter throughout the northern hemisphere, including northern China, North America and Northern Europe [3]. In the case of Inner Mongolia, the region of this study, ambient temperatures are mostly below 0 °C and can easily be as low as −20 °C or even lower from mid December to mid February, which makes sheep and, in particular, lambs readily subject to cold stress since they are exposed to a cold environment. To ensure that lactating ewes utilize the peak production of fodder during the winter rainfall period, pregnancy in sheep commonly occurs from late autumn to midwinter (the months from August to September) and lambing takes place from January to February. Therefore, lambs might be born in challenging circumstances and exposed to cold stress by compromising lamb survival [18,19]. As compared to adults, the peculiarities of thermoregulatory control such as an immature central nervous system and a ratio of body surface to body weight have been observed in individuals in the early postnatal period including weaner lamb, which can lead to young animals being more affected by cold stress [13]. It is already established that a thermal (cold and heat) environment can cause stress in lamb and bring about physiological changes. Profound negative influences have been reported in lamb when under prolonged exposure to even mildly low ambient temperatures, including alterations in voluntary dry matter (DM) intake and DM digestibility, basal metabolic rate, hormonal status, immunity, wool yield, methane yield, and body weight [20,21]. Peana et al. [22] performed a relation analysis between milk production of Sardinian sheep and meteorological conditions in winter and early spring, and found that milk yield decreased by 25% (0.30 kg/d) in Sarda sheep when temperature decreased from the optimal value of 9–12 °C to −3 °C, and with a progressive decrease in milk yield as the temperature decreased. A more significant finding by Ramon et al. [23] for Manchega dairy sheep is that the decline in productivity due to cold stress was greater than that due to heat stress, especially for milk volume. What that means is that the effects of cold stress on animals might be greater than that of heat stress.

Subjecting an organism to cold can cause an increase in the activity of the sympathoadrenal system and rescaling of the whole metabolic machinery, a protective adaptation mechanism of homoiotherm taken to ensure the constant temperature of the body [13]. The process results in much more ROS. A low ROS concentration is beneficial to the host through the immune response, while excessive production of ROS can cause oxidative stress, damage the host and cause inflammation in both ruminants and monogastric animals [24]. Remarkably, recent results of studies on the effects of cold stress on immune function in animals are inconsistent. A study found that cold stress enhanced antibody production [25], whereas Guo et al. [26] stated that the immunoglobulin G (IgG)- and Th1-related cytokines in the blood were significantly lower when sheep were exposed to cold environments and/or stronger wind. The difference may be concerned with the duration and severity of cold stress treatment as well as the acclimation level of the animals.

Exploring how stressors affect the immune and the antioxidative system of lambs is highly relevant to establish stress mitigation strategies that are more appropriate for ovine production systems. However, little is known about the effect of cold environments on growth, antioxidant status and immunity in sheep, especially lambs. In this sense, the present study aims to examine the variation in and possible mechanism of antioxidant and immune indexes and related gene expression in lambs in cold environment conditions.

## 2. Materials and Methods

### 2.1. Ethical Approval

This study was conducted after approval by the Animal Care and Use Committee of Inner Mongolia Agricultural University and was performed following the national standard Guideline for Ethical Review of Animal Welfare (GB/T 35892-2018). All efforts were made to minimize the suffering of animals.

### 2.2. Animal and Experiment Design

The 28 day experiment was conducted in the sheep experimental farm of Inner Mongolia Agricultural University, Hohhot, Inner Mongolia, China (Location: 40°51′~41°8′ North Latitude, 110°46′~112°10′ East Longitude), during winter. Eighteen five-month-old healthy Dorper × Mongolian crossbred male lambs, with an average body weight of 33.07 ± 2.17 kg and a fleece thickness of 40 mm, were randomly selected from the farm. They were randomly divided into 3 groups, with 6 lambs in each group. The experimental lambs of different groups were fed in different sheds, and each lamb was fed in a single pen. The IHC group (indoor heating control group) was in a house with a comfortable thermal environment controlled with electric heating; the IT group (indoor temperature group) was in a house without heating, a natural indoor cold environment in winter, with intermittent and slight degrees of stimulation of coldness; the OT group (outdoor temperature group) was exposed to a cold environment in an external natural environment, only equipped with a sunshade net to decrease solar radiation. Due to the low ambient temperature, water was changed every 2 h to ensure that the lambs could drink water ad libitum. Lambs in all three groups received (eat ad libitum and ensure 500 g of residual feed per time) the same total mixed ration of granulated feeds twice a day (at 08:00 and 15:00). The diet of the experimental lambs was formulated according to the nutritional requirements and the physiological state of the sheep (NRC 2007).

### 2.3. Sample Collection

The experiment was conducted over 28 days. On days 14 and 28, jugular vein blood samples were collected into both vacutainers without anticoagulant and vacutainers with ethylene diamine tetra acetic acid (EDTA) from empty stomachs from 07:00 to 08:00. These blood samples were centrifuged for 20 min at 3000× *g* to collect serum and leukocytes, respectively. The serum samples were frozen at −20 °C for analysis of antioxidative and immune indexes, and the leukocytes were stored at −80 °C for total RNA extraction and gene expression analysis.

### 2.4. Measurement

#### 2.4.1. Environmental Parameters

Data on temperature in the barn were monitored and recorded continuously with sensors (ZJI-2A, Shanghai Longtuo Instrument and equipment Co., Ltd., Shanghai, China), which were used for calculating hourly means of temperature throughout the experimental period. One temperature sensor was also placed outside the barn. The wind speed in the three treatments was measured at 08:00, 14:00 and 20:00 daily by a small vane anemometer (testo 416-Small vane anemometer, Testo SE & Co. KGaA, West Chester, PA, USA), at the center of the barn 1 m above the ground.

#### 2.4.2. Growth Performance

Feed offered and refused was weighed and recorded daily to calculate the average daily feed intake (ADFI). Lambs were weighed prior to the morning feeding on day 1, 14, and 28 of the trial. The average daily gain (ADG) and feed-to-gain ratios (F:G) were calculated for each lamb.

#### 2.4.3. Determination of Antioxidative and Immune Parameters in Serum

Total antioxidant capacity (T-AOC) and malondialdehyde (MDA) content and the activity of enzymes including total superoxide dismutase (T-SOD), glutathione peroxidase (GPx), and catalase (CAT) in serum were measured by the spectrophotometric method according to the instructions of the commercial kits (Nanjing Jiancheng Institute of Bioengineering, Nanjing, China).

The concentration of interleukins 1β (IL-1β), interleukins 2 (IL-2), tumor necrosis factor α (TNF-α), immunoglobulin A (IgA), IgG, and immunoglobulin M (IgM) in serum was analyzed using lamb-specific ELISA kits (Ruixin Biological Technology Co., Ltd. Quanzhou, China) following the manufacturer’s instruction.

#### 2.4.4. Total RNA Extraction and Quantitative RT-PCR Analysis

Total RNA in the leukocytes was extracted using Trizol reagent (Invitrogen, Carlsbad, CA, USA). The purity and concentration of isolated RNA were determined from the ratio of absorbance at 260 and 280 nm using a NanoDrop spectroscopy (NanoDrop Technologies, Wilmington, DE, USA) and adjusted to the same concentration (500 ng/μL). RNA integrity was analyzed using horizontal electrophoresis through 1.5% agarose gel (Appendix A). Two microliters (μL) of total RNA (500 ng/μL) from each sample was reverse-transcribed into cDNA using the PrimeScript RT Reagent Kit after removing genomic DNA contamination according to the manufacturer’s instructions (Takara Bio Inc., Otsu, Japan). Quantification of cDNA transcript was performed using qPCR TB Green Kit with the gene-specific primer on the LightCycler 96 real-time PCR system. The qRT-PCR was performed using 20 μL reactions, with cycling conditions as described previously [27]. A subsequent dissociation stage produced a melting curve to check and verify the specificity and purity of the amplified products. All samples were run in triplicate, and the average values were obtained. The optimum annealing temperatures for different genes (Table 1) were designed and synthesized by TaKaRa Biotechnology Co., Ltd., Dalian, China. The abundance of β-actin mRNA was not influenced by different environmental treatments. The relative transcript quantities of target genes in the mRNA expression level were calculated using the 2^−ΔΔCT^ method after normalization to the reference gene, β-actin [28]. The values of the IHC group were used as a calibrator. The qRT-PCR amplification efficiency of each gene was calculated based on the slope of the log_10_ diluted cDNA relative standard curve generated by pooled samples, and it was found that the amplification efficiency values were consistent between the target genes and the reference gene. The correlation coefficients (r^2^) of all standard curves were >0.99 and the amplification efficiency was between 90% and 110%, allowing the use of the 2^−ΔΔCT^ method for the calculation of relative gene expression levels.

### 2.5. Statistical Analysis

Data were managed using Microsoft Excel 2017 and analyzed by one-way ANOVA with the GLM procedure of SAS (Version 9.2, SAS Institute Inc., Cary, NC, USA) with the individual lamb as the experimental unit. The statistical model used for the analysis was: Yij = μ + Ai + Eij, where Yij = dependent variable; μ = overall mean, Ai = effect of the degree of the cold environment (range = 1 to 3), and Eij = random effect. The results were expressed as the least squares mean and standard error of means (SEM). Tukey’s multiple range test was used to compare the mean values (*p* < 0.05) to show the significant differences.

## 3. Results

### 3.1. Environmental Conditions

The average hourly ambient temperature of the three treatments was monitored and calculated during the 28-day experiment period. As shown in Figure 1, for the IHC group, the daily average temperature was 8.28 °C (ranging from 6.95 to 10.92 °C). For the IT group, the daily average temperature was 1.82 °C (ranging from −0.41 to 6.30 °C). For the OT group, the daily average temperature was −13.05 °C (ranging from −21.91 to 0.99 °C). In addition, the average daily ambient temperature of each group is shown in Table 2, showing that the temperature of the IHC group was the highest and that of the OT group was the lowest.

The wind speed of different treatments at 08:00, 14:00 and 20:00 during the test period are shown in Table 3. The wind speed and variation in amplitude of the indoor treatment (IHC and IT groups) were small, showing that the wind speed of the IHC group was 0.03–0.04 m/s, and that of the IT group was 0.04 m/s. The wind speed and variation in amplitude of the OT group was the greatest, ranging from 0.67 to 1.08 m/s.

### 3.2. Growth Performance

As shown in Table 4, the ADFI of lambs in the OT group was significantly higher than that in the IHC and IT groups on d 1 to 14 (*p* < 0.05). The ADG in the IT and OT groups was significantly lower than that in the IHC group on d 15 to 28 (*p* < 0.05). The F:G in the OT group was significantly higher than that in the IHC group (*p* < 0.05), but there was no significant difference between the IT group and the OT group (*p* > 0.05). During the whole experiment period (d 1 to 28), the ADG in the OT group was significantly lower than that in the IHC group, while the ADFI and the F:G in the OT group were significantly higher than those in the IHC group (*p* < 0.05). On d 1, 14, and 28, there was no difference in body weight among the three groups (*p* > 0.05).

### 3.3. Immune Indexes and Relevant Genes Expression

The concentrations of serum Igs and cytokines are shown in Table 5. As described in Table 5, on d 14, there was no significant difference in serum immune indexes among all treatments (*p* > 0.05). On d 28, IgA and IL-2 contents in serum of lambs in the IT and OT groups were significantly lower than those in the IHC group, whereas the content of TNF-α in the OT group was significantly higher than that in the other groups (*p* < 0.05). There were no significant differences in IgG, IgM, and IL-1β among the three groups (*p* > 0.05).

As shown in Figure 2, on d 14, there was no significant difference in the relative expression levels of immune-related genes among all groups (*p* > 0.05). On d 28, compared with the IHC group, the relative mRNA expression level of IL-2 of lambs in the IT and OT groups was significantly decreased, whereas the relative mRNA expression levels of TNF-α and nuclear factor kappa B p65 (NF-κB p65) were significantly increased in the OT group (*p* < 0.05). The relative mRNA expression levels of IL-1β and NF-κB p50 were not significantly different among all groups (*p* > 0.05).

### 3.4. Antioxidative Properties and Relevant Genes Expression

The results of antioxidant enzyme activities, total antioxidant capacity, and MDA content in serum were displayed in Table 6 and the relevant gene expression results were presented in Figure 3. As described in Table 6, on d 14, the GPx and T-SOD activities of lambs in the OT group were significantly lower than those in the other groups (*p* < 0.05). T-AOC of lambs in the IT and OT groups was significantly lower than that in the IHC group (*p* < 0.05). There was no significant difference in other indexes among groups (*p* > 0.05). On d 28, the activities of CAT, GPx, and T-SOD in the IT and OT groups were significantly lower than those in the IHC group, and the activity of GPx and T-SOD in the OT group were significantly lower than those in the IT group (*p* < 0.05). T-AOC of lambs in the OT group was significantly lower than that in the other groups, while MDA content was significantly higher than that in the IHC group (*p* < 0.05).

As shown in Figure 3, on day 14, the relative mRNA expression level of SOD1 of lambs in the IT and OT groups was significantly lower than that in the IHC group, but the relative mRNA expression levels of GPx and nuclear factor-erythroid 2-related factor 2 (Nrf2) in the OT group were significantly higher than those in the IHC group (*p* < 0.05). There was no significant difference in the relative expression levels of other genes among groups (*p* > 0.05). On day 28, the relative mRNA expression levels of CAT, GPx and SOD2 in the IT and OT groups were significantly lower than those in the IHC group, and the relative mRNA expression levels of CAT and SOD2 in the OT group were significantly lower than those in the IT group (*p* < 0.05). The relative mRNA expression levels of SOD1 and Nrf2 in the OT group were significantly lower than those in the IHC group (*p* < 0.05).

## 4. Discussion

In northern China, the winter temperature is generally low, and the cold period is generally long. This kind of long-term cold stimulation and sudden snow, cold weather and other temporary strong cooling conditions have a great impact on the growth and body health of animals, reducing the productivity and disease resistance of livestock. In addition, sustained low temperatures are often accompanied by strong winds, which speed up the process of heat loss when the ambient temperature is lower than the animal’s surface temperature. The constant circulation of cold air continually removes the heat of the animal itself [29]. In the current study, the temperature and wind speed of indoor treatment groups (IHC and IT groups) were relatively stable, showing that the average temperature of the IHC group and the IT group was 8.28 and 1.82 °C, respectively, over the whole period. In addition, the wind speed of the two indoor treatments was low, ranging from 0.03 to 0.04 m/s. In contrast, the average temperature of the OT group was −13.05 °C over the whole period. In addition, the average temperature of the OT group was −15.49 °C from d 15 to 28, which was 4.87 °C lower than that of −10.62 °C from d 1 to 14. The wind speed and variation in amplitude of the OT group were greatest, and the maximum average wind speed was 1.36 m/s, which indicated that the lambs in the OT group experienced the most severe cold stimulation.

In cold environments, to maintain body temperature, the basal metabolism is increased, catabolism is enhanced, and energy deposition is reduced. Moreover, cold environments cause gastrointestinal contraction and peristalsis of animals, shorten the existence time of chyme in the intestinal tract, and increase the gastric emptying rate, leading to increased appetite and thus increased feed intake [30]. However, cold environments can also cause a decrease in the reticulum volume of ruminants, which empties the digesta without being thoroughly digested and absorbed, resulting in a decrease in the apparent digestibility of feed [31]. Therefore, cold environments can reduce the feed utilization rate of animals, and cause a decrease in animal weight gain. In a certain range of low temperatures, the digestibility of sheep was positively correlated with the ambient temperature [32]. The dry matter digestibility of sheep decreased with decreasing temperature [14]. A related study found that the daily gain of 2-month-old lambs was reduced after cold stimulation [33]. The apparent digestibility of dry matter, calcium and phosphorus and daily gain of 6-month-old lambs were significantly reduced after cold stimulation [34]. Moreover, calf feed consumption increased in cold environments, but there was no change in body weight [35]. The apparent digestibility of crude protein and ether extract of pigs decreased and feed consumption increased in cold environments, but there was no change in body weight [36,37]. In line with previous studies, our research found that the ADFI and the F:G increased while the ADG decreased in lambs in cold environments. It was noted that on d 15 to 28 of this experiment, there was no difference in the ADFI among all groups, which might be related to the fact that the feed intake of lambs recovered due to their own regulation with extended time in cold environments. However, the digestibility of lambs may not have increased due to the recovery of feeding of lambs, thereby resulting in a lower F:G.

Furthermore, in cold environments, the structure of the rumen in ruminants determines that it only promotes mixing, sorting, and fluid pushing of small pellet feed. Therefore, the negative effect of cold environments on pellet feed digestibility was significantly stronger than that of chopped forage [38]. In the present study, lambs were fed pellet feed, which partly explained the decrease in nutrient apparent digestibility of lambs in cold environments.

Although the conclusions about how cold environments affect the immune function of animals are inconsistent, there is widespread agreement that the immune response can be suppressed in cold-adapted animals. A study of cold stress in rats showed that both acute cold stress and long-term chronic cold stress inhibited the immune response of rats, and long-term chronic cold stress had a stronger inhibitory effect on the immune function of the organism [39]. As one of the indicators reflecting humoral immunity function, serum immunoglobulins can combine with antigens to exert a variety of biological effects and stimulate antibody production [40]. Among them, IgG is the main immunoglobulin, which can promote immune cells to phagocytose pathogens and neutralize bacterial toxins. IgA plays a remarkable protective and defensive role against invading pathogens, and its content in serum is second only to that of IgG. Chen et al. [41] found that serum IgA and IgG contents in Simmental cows under chronic cold stress in winter were significantly lower than those in cows under non-stress conditions (in autumn). This finding was similar to ours, in which the IgA content in the serum of lambs in the IT and OT groups was significantly lower than that in the IHC group, suggesting that cold environments might inhibit humoral immune function by affecting the synthesis of immunoglobulins.

The content of immune cytokines in blood is one of the surest indicators of cellular immune function. When animals are stimulated by stressors, macrophages can produce and secrete a large number of inflammatory cytokines, including IL-1β, IL-6 and TNF-α, which mediate the inflammatory response of animals. The anti-inflammatory cytokine IL-2 and the proinflammatory cytokine TNF-α are secreted by T lymphocyte subtypes Th1 and Th2 cells, respectively. As an important anti-inflammatory cytokine, IL-2 is secreted by mitogen- or specific antigen-stimulating lymphocytes, mainly promoting the growth of T cells, stimulating the proliferation and differentiation of B cells and producing antibodies, and mediating the cellular immune response [42]. TNF-α, as the main proinflammatory factor in the early inflammatory response, can bind to a variety of receptors on immune cells and thus plays an immunoregulatory role [43]. However, with the increase in TNF-α secretion, the inflammatory response is further intensified, and eventually causes tissue damage and reduces immune function [43]. In the present study, compared with the IHC group, cold environments significantly decreased the IL-2 content and the mRNA expression level of lambs in the IT group and the OT group on d 28, while the inflammatory factor TNF-α content and the mRNA expression level of lambs in the OT group were significantly increased. Our experimental results were consistent with those of previous studies. Yang et al. [44] found that serum IL-2 content showed a decreasing trend and IL-4 content decreased significantly in Altay sheep at −15 to −30 °C. Shini et al. [45] found that chicken immune function was inhibited after cold stress, and the TNF-α content in peripheral leukocytes was significantly increased. Combined with these results and based on our study, it is reasonable to infer that cold environments can reduce or increase the secretion of anti-inflammatory cytokine (IL-2) and proinflammatory cytokine (TNF-α), respectively, by affecting their gene expression levels, resulting in the dysregulation of the Th1/Th2 cytokine profile disruption of the Th1/Th2 balance, thus affecting the proliferation and differentiation of T lymphocytes and B lymphocytes, causing immune imbalance, inducing inflammatory responses and eventually inhibiting the immune function of the organism. Ultimately, the immune function of lambs was suppressed. NF-κB is a key transcription factor associated with the immune response. Under physiological conditions, functional NF-κB dimers combine with the inhibitor protein of NF-κB (IκB) in the cytosol and act in an inactive form [46]. Under low-temperature stimuli, NF-κB dissociates from IκB, which is triggered by another kinase, IκB kinase (IKK), and subsequently translocates into the nucleus, where it upregulates the expression of proinflammatory cytokines, including IL-6 and TNF-α, eventually exaggerating the initial inflammatory responses [46]. The results of our study showed that the mRNA expression of NF-κB p65 was significantly increased in the OT group on d 28, which might be due to the activation of NF-κB into the nucleus by excessive ROS production in cold environments. Moreover, the increased expression and secretion of TNF-α, the downstream target of NF-κB, observed in the OT group could further illustrate this point. The main physiological response of animals in cold environments is to generate heat and maintain a constant body temperature. Consequently, dietary nutrients and energy are diverted from maintaining the stability of the immune system to thermoregulation, which may be one of the reasons why cold stress inhibits immune function [47].

It has been proven that extreme heat and cold environments can cause redox imbalance, producing oxidative stress that affects the health and growth of animals [48,49]. In addition, to maintain a constant body temperature under cold stress, the metabolic rate of the body will increase, and ROS production will also increase with the enhancement of mitochondrial energy metabolism [6]. ROS, which are produced in excess and cannot be scavenged in time, can oxidize unsaturated fatty acids to produce MDA and cause oxidative damage to other biomolecules (protein and DNA/RNA) in many organs and tissues, ultimately leading to many neurodegenerative and inflammatory diseases [50]. Antioxidant enzyme systems, such as CAT, SOD, and GPx, play an important role in alleviating oxidative stress in animals. Among them, SOD catalyzes the dismutation of harmful superoxide anions into molecular oxygen followed by conversion into hydrogen peroxide, and CAT catalyzes the decomposition of hydrogen peroxide in the forms of harmless water and oxygen [51]. The T-AOC level represents the cumulative action of all antioxidants, including enzymatic and non-enzymatic (ascorbic acid, α-tocopherol, β-carotene, etc.) compounds, hence providing a comprehensive parameter that reflects the total antioxidant capacity of organisms rather than the simple sum of measurable antioxidants [52]. MDA is one of the most common metabolites produced during lipid peroxidation of biological membranes and causes cellular dysfunction, as well as an established indicator of oxidative stress [53]. In the present study, we found that cold environments significantly reduced the activities of GPx and T-SOD and the content of T-AOC and downregulated the mRNA expression of SOD1 in the OT group lambs on day 14. In addition, on day 28, the activities of CAT, GPx, and T-SOD and the content of T-AOC in the IT group and the OT group were significantly decreased; the mRNA expression of CAT, GPx, SOD1, and SOD2 in the OT group was downregulated; and the MDA content in the OT group was significantly increased. These results suggest that cold environments might inhibit the activity of antioxidant enzymes by downregulating the gene expression of antioxidant enzymes and thus accelerate lipid peroxidation in the cells and tissues of lambs. The increased MDA level observed in the OT group on day 28 could be good evidence of this point. A similar result was reported for lambs by Wang et al. [54], who found that cold stress decreased plasma GPx and CAT activities and the T-AOC content while increasing the MDA content, and the antioxidant function was damaged. Furthermore, another study also indicated that chronic cold stress reduces the activity of CAT, GPx, and SOD in chicken heart tissue [55]. The Nrf2 pathway is a key transduction pathway to resist oxidative stress. Nrf2 can regulate the expression of downstream representative target genes involved in the antioxidation process, such as SOD, CAT, and GPx, which have a great effect on the elimination of excessive ROS [56]. Under physiological conditions, Nrf2 binds to its inhibitor Keap1 in the cytoplasm and acts in an inactive form [56]. However, under oxidative stress, Keap1 is modified to release Nrf2 from the cytoplasm into the nucleus, which initiates the transcription and translation of a series of downstream protective genes, such as antioxidant proteins and phase II detoxification enzymes [56]. In this study, the relative mRNA expression of Nrf2 of lambs in cold environments was measured. The results showed that the mRNA expression of Nrf2 was significantly increased on day 14 but significantly decreased on day 28. In addition, our study found that the mRNA expression of GPx in the OT group was significantly higher than that in the IHC group, while the enzyme activity was significantly lower than that in the IHC group on day 14. This result may be due to the oxidative stress caused by cold environments in the early stage of the experiment, and the excess ROS activated Nrf2 into nuclear translation, thus upregulating the expression of downstream GPx; however, with the increase in the severity and duration of cold stress, a large amount of GPx, CAT, and SOD was consumed by the accumulated ROS, resulting in a decrease in antioxidant enzyme activity. In addition, the variation in the relative gene expression of GPx (d 14) was not totally in line with the variation in the enzymatic activities, which might be due to the enzyme activity being affected by many factors, not just gene expression. In addition, another study showed that NF-κB p65 could decrease the DNA-binding activity of Nrf2, leading to the inhibition of the Nrf2 signaling pathway [57]. This may be one of the reasons for the inhibition of Nrf2 expression and the reduction in antioxidant enzyme activity by cold stress on day 28.

## 5. Conclusions

The results obtained from this study showed that lambs reared outdoors in winter showed lower growth performance, which was primarily reflected in the increased ADFI, a decreased ADG, and an increased F:G. Moreover, lambs reared outdoors in winter had a worse situation in terms of immunity and antioxidant status due to low temperature and stronger wind. However, shelter feeding in winter reduced the adverse impact of cold environments on lambs, and housing with heating equipment was more conducive to improve the growth, immune, and antioxidant function of the lambs.

## Figures and Tables

**Figure 1 animals-12-02535-f001:**
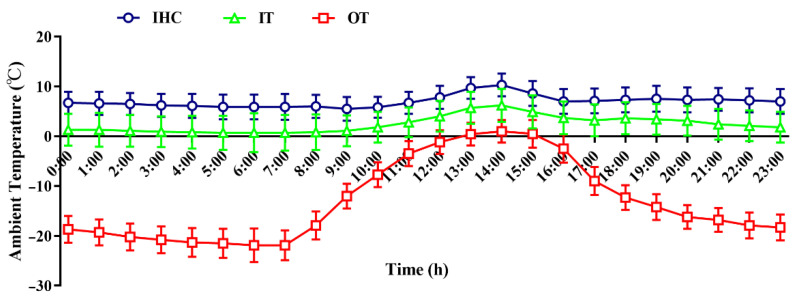
Daily variation in the ambient temperature in different groups during the experimental period. Note: Each value is shown as the mean ± standard deviation (SD) (*n* = 28). IHC group, indoor heating control group; IT group, indoor temperature group; OT group, outdoor temperature group.

**Figure 2 animals-12-02535-f002:**
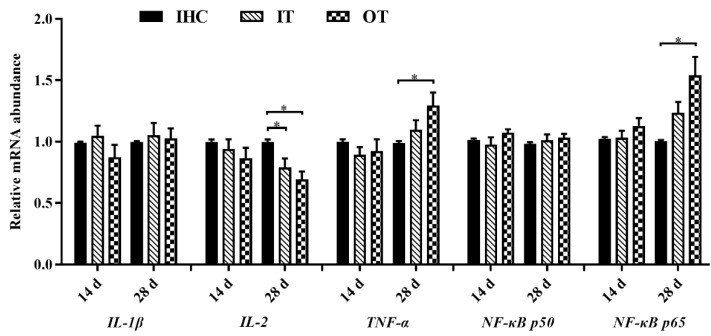
Effects of cold environments on the relative gene expression level of *IL-1β*, *IL-2*, *TNF-α*, *NF-κB p50*, and *NF-κB p65*. Note: Each value is shown as the mean and SEM (*n* = 6). Data columns with * mean a significant difference between groups (*p* < 0.05). *IL-1β*, interleukin 1 beta; *IL-2*, interleukin 2; *TNF-α*, tumor necrosis factor alpha; *NF-κB p50*, nuclear factor kappa B p50; *NF-κB p65*, nuclear factor kappa B p65. IHC group, indoor heating control group; IT group, indoor temperature group; OT group, outdoor temperature group.

**Figure 3 animals-12-02535-f003:**
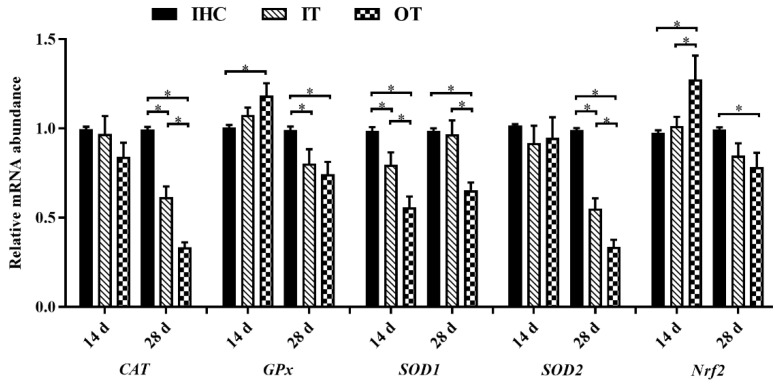
Effects of cold environments on the relative gene expression level of *CAT*, *GPx*, *SOD1*, *SOD2*, and *Nrf2*. Note: Each value is shown as the mean and SEM (*n* = 6). Data columns with * mean a significant difference between groups (*p* < 0.05). *CAT*, catalase; *GPx*, glutathione peroxidase; *SOD1*, Cu/Zn-superoxide dismutase; *SOD2*, Mn-superoxide dismutase; *Nrf2*, nuclear factor-erythroid 2-related factor 2; IHC group, indoor heating control group; IT group, indoor temperature group; OT group, outdoor temperature group.

**Table 1 animals-12-02535-t001:** A list of primers in qRT-PCR analysis of mRNA expression of the target genes.

Target	GenBank Accession	Sequence of Nucleotide (5′ to 3′)	Annealing Temp (°C)	Fragment Size (bp)
*SOD1*	NM_001145185	For: GGAGACCTGGGCAATGTGAA Rev: CCTCCAGCGTTTCCAGTCTT	60	182
*SOD2*	NM_001280703.1	For: AAACCGTCAGCCTTACACC Rev: ACAAGCCACGCTCAGAAAC	60	116
*GPx*	XM_004018462.1	For: TGGTCGTACTCGGCTTCCC Rev: AGCGGATGCGCCTTCTCG	60	163
*CAT*	XM_004016396	For: GAGCCCACCTGCAAAGTTCT Rev: CTCCTACTGGATTACCGGCG	60	148
*Nrf2*	XM_004004557.1	For: TGTGGAGGAGTTCAACGAGC Rev: CGCCGCCATCTTGTTCTTG	61	88
*IL-1β*	NM_001009465	For: CGATGAGCTTCTGTGTGATG Rev: CTGTGAGAGGAGGTGGAGAG	59	161
*IL-2*	AF215687.1	For: AATTGAGCTTAGGCGTATCTACAGG Rev: TACTCGTCTTGGCTTCATTCACA	61	80
*TNF-α*	NM_001024860	For: AGTCTGGGCAGGTCTACTTTG Rev: GGTAACTGAGGTGGGAGAGG	60	127
*NF-κB p50*	XM_004009667.3	For: AGCACCACTTATGACGGAACTACA	60	168
		Rev: GACCCCTTCATCCTCTCCATC		
*NF-κB p65*	XM_004020143.3	For: GGAGGCCAAGGAACTGAAGA	60	101
		Rev: TCAGGGGCAGAGAGAAGGAG		
*β-actin*	NM_001009784.1	For: GAGCGCAAGTACTCCGTGTG Rev: CATTTGCGGTGGACGATG	58	122

Note: *SOD1*, Cu/Zn-superoxide dismutase; *SOD2*, Mn-superoxide dismutase; GPx, glutathione peroxidase; *CAT*, catalase; *Nrf2*, nuclear factor-erythroid 2-related factor 2; *IL-1β*, interleukin 1 beta; *IL-2*, interleukin 2; *TNF-α*, tumor necrosis factor alpha; *NF-κB p50*, nuclear factor kappa B p50; *NF-κB p65*, nuclear factor kappa B p65; *β-actin*, beta actin; For, forward primer; Rev, reverse primer.

**Table 2 animals-12-02535-t002:** The average daily ambient temperature in different groups during the experimental period (°C).

Items	Groups			Items	Groups		
	IHC	IT	OT		IHC	IT	OT
Day 1	4.98 (3.12, 8.21)	2.10 (−0.21, 8.04)	−8.33 (−18.51, 7.74)	Day 15	7.11 (4.43, 13.91)	5.43 (0.04, 15.18)	−17.19 (−27.08, −1.07)
Day 2	6.09 (3.37, 12.74)	2.66 (0.27, 8.85)	−9.68 (−18.52, 5.22)	Day 16	6.64 (3.70, 9.91)	0.46 (−3.22, 3.76)	−15.70 (−28.06, 3.25)
Day 3	6.30 (4.01, 12.13)	2.13 (0.16, 7.82)	−10.18 (−18.09, 5.01)	Day 17	7.62 (5.97, 13.04)	1.55 (−0.53, 5.10)	−13.53 (−23.86, 2.21)
Day 4	6.99 (4.75, 13.21)	2.47 (−0.25, 8.01)	−11.77 (−21.65, 2.64)	Day 18	6.72 (3.33, 10.21)	2.97 (0.48, 6.90)	−14.20 (−24.29, −0.67)
Day 5	7.33 (5.02, 12.16)	3.10 (1.34, 8.89)	−10.70 (−20.09, 4.93)	Day 19	6.34 (3.96, 11.87)	2.63 (0.68, 7.39)	−17.09 (−28.80, −1.52)
Day 6	7.96 (6.13, 12.73)	4.69 (2.03, 10.94)	−6.83 (−18.19, 10.93)	Day 20	6.38 (4.41, 9.53)	2.18 (−0.43, 4.31)	−15.03 (−28.04, −1.01)
Day 7	9.53 (8.06, 11.35)	4.26 (1.97, 7.54)	−2.06 (−11.22, 12.10)	Day 21	2.58 (−1.38, 5.52)	−0.05 (−2.47, 2.18)	−15.53 (−26.31, −7.68)
Day 8	10.79 (6.91, 12.55)	3.70 (1.80, 5.31)	−1.33 (−14.00, 2.07)	Day 22	4.62 (2.55, 9.03)	0.28 (−1.11, 2.63)	−19.11 (−27.82, −5.01)
Day 9	6.27 (2.93, 8.21)	3.69 (1.10, 8.29)	−12.98 (−21.67, −1.05)	Day 23	8.87 (6.68, 11.02)	1.21 (−0.33, 4.28)	−15.87 (−26.51, −3.05)
Day 10	6.47 (3.20, 13.32)	2.89 (0.13, 12.91)	−13.80 (−23.20, −2.31)	Day 24	8.28 (2.77, 11.11)	0.79 (−2.52, 2.76)	−17.61 (28.20, 0.02)
Day 11	6.95 (2.15, 14.37)	2.76 (−0.65, 12.32)	−15.22 (−26.69, −3.24)	Day 25	11.20 (8.04, 17.93)	2.40 (−0.60, 7.20)	−13.59 (−26.10, 6.05)
Day 12	4.68 (0.63, 6.97)	−1.09 (−6.10, 4.52)	−17.38 (−27.85, −3.94)	Day 26	11.55 (9.21, 15.02)	2.89 (−0.18, 6.84)	−10.13 (−23.53, 10.54)
Day 13	4.07 (0.82, 5.78)	3.90 (0.03, 9.01)	−17.77 (−27.71, −3.84)	Day 27	9.04 (6.09, 12.67)	1.50 (−0.43, 4.71)	−15.78 (−28.35, −1.09)
Day 14	5.51 (2.16, 7.68)	3.93 (−0.53, 6.08)	−16.81 (−26.84, −1.82)	Day 28	9.05 (6.27, 12.80)	0.75 (−1.53, 2.75)	−15.42 (−24.76, −4.70)

Note: Each value is shown as the average, minimum and maximum temperatures for each day. IHC group, indoor heating control group; IT group, indoor temperature group; OT group, outdoor temperature group.

**Table 3 animals-12-02535-t003:** Change in wind speed in different groups during the experimental period (m/s).

Items	Groups			*p*-Value
	IHC	IT	OT	
08:00	0.03 ± 0.01 ^b^	0.04 ± 0.02 ^b^	0.67 ± 0.65 ^a^	<0.001
14:00	0.04 ± 0.01 ^b^	0.04 ± 0.02 ^b^	1.08 ± 0.89 ^a^	<0.001
20:00	0.03 ± 0.01 ^b^	0.04 ± 0.01 ^b^	0.80 ± 1.11 ^a^	<0.001

Note: Each value is shown as the mean ± SD (*n* = 28). In the same row, values with no letter (a and b) or the same letter superscripts mean no significant difference (*p* > 0.05), while those with different letter superscripts mean a significant difference (*p* < 0.05). IHC group, indoor heating control group; IT group, indoor temperature group; OT group, outdoor temperature group.

**Table 4 animals-12-02535-t004:** Effects of cold environments on growth performance of lambs.

Items	Groups			SEM	*p*-Value
	IHC	IT	OT		
BW (kg)					
1 d	33.04 ± 2.07	33.07 ± 2.43	33.10 ± 2.40	0.88	0.999
14 d	35.16 ± 2.43	35.09 ± 2.46	35.11 ± 2.33	0.92	0.999
28 d	38.17 ± 2.64	37.49 ± 2.42	37.29 ± 1.95	0.92	0.797
ADG (g/d)					
1–14 d	151.90 ± 27.40	144.29 ± 45.61	144.05 ± 32.03	13.83	0.912
15–28 d	215.00 ± 28.82 ^a^	171.19 ± 33.70 ^b^	155.71 ± 34.72 ^b^	16.33	0.017
1–28 d	183.45 ± 22.99 ^a^	157.74 ± 19.06 ^ab^	149.88 ± 28.04 ^b^	10.89	0.065
ADFI (g/d)					
1–14 d	1380.2 ± 49.4 ^b^	1383.8 ± 33.1 ^b^	1436.7 ± 15.5 ^a^	32.67	0.025
15–28 d	1549.1 ± 48.5	1586.3 ± 51.6	1597.1 ± 21.0	37.04	0.159
1–28 d	1464.6 ± 39.7 ^b^	1486.9 ± 39.2 ^ab^	1516.9 ± 15.3 ^a^	24.1	0.050
F:G					
1–14 d	9.31 ± 1.49	10.47 ± 3.36	10.36 ± 2.10	2.13	0.669
15–28 d	7.33 ± 1.19 ^b^	9.60 ± 2.05 ^ab^	10.75 ± 2.67 ^a^	1.79	0.034
1–28 d	8.09 ± 1.04 ^b^	9.53 ± 1.08 ^ab^	10.45 ± 2.14 ^a^	1.32	0.048

Note: Each value is shown as the mean ± SD (*n* = 6). In the same row, values with no letter (a and b) or the same letter superscripts mean no significant difference (*p* > 0.05), while those with different letter superscripts mean a significant difference (*p* < 0.05). BW, body weight; ADG, average daily gain; ADFI, average daily feed intake; F:G, feed to gain ratio. IHC group, indoor heating control group; IT group, indoor temperature group; OT group, outdoor temperature group; SEM, standard error of means.

**Table 5 animals-12-02535-t005:** Effects of cold environments on immune indexes of lambs.

Items	Groups			SEM	*p*-Value
	IHC	IT	OT		
14 d					
IgA (μg/mL)	51.38 ± 6.57	44.85 ± 10.09	44.95 ± 9.73	3.63	0.443
IgG (μg/mL)	100.14± 14.17	119.10 ± 24.69	117.33 ± 11.62	7.70	0.159
IgM (μg/mL)	253.76 ± 38.64	274.78 ± 27.86	252.79 ± 28.51	13.07	0.445
IL-1β (pg/mL)	219.71 ± 20.10	255.59 ± 62.19	203.90 ± 19.57	18.19	0.162
IL-2 (pg/mL)	271.87 ± 20.74	260.74 ± 65.76	240.66 ± 29.66	17.83	0.511
TNF-α (pg/mL)	33.18 ± 4.04	32.95 ± 6.75	35.44 ± 6.60	2.29	0.752
28 d					
IgA (μg/mL)	59.26 ± 6.31 ^a^	45.87 ± 3.83 ^b^	39.37 ± 4.42 ^b^	3.85	<0.001
IgG (μg/mL)	114.46 ± 34.41	124.43 ± 20.87	114.70 ± 21.90	10.41	0.791
IgM (μg/mL)	240.63 ± 42.93	282.86 ± 35.93	274.12 ± 29.08	15.77	0.196
IL-1β (pg/mL)	233.11 ± 35.61	282.55 ± 63.32	251.00 ± 49.67	21.35	0.268
IL-2 (pg/mL)	305.53 ± 52.64 ^a^	237.00 ± 35.52 ^b^	216.06 ± 47.07 ^b^	22.90	0.042
TNF-α (pg/mL)	37.05 ± 10.00 ^b^	39.57 ± 5.60 ^b^	53.27 ± 9.74 ^a^	4.48	0.022

Note: Each value is shown as the mean ± SD (*n* = 6). In the same row, values with no letter (a and b) or the same letter superscripts mean no significant difference (*p* > 0.05), while those with different letter superscripts mean a significant difference (*p* < 0.05). IgA, immunoglobulin A; IgG, immunoglobulin G; IgM, immunoglobulin M; IL-1β, interleukin 1β; IL-2, interleukin 2; TNF-α, tumor necrosis factor alpha. IHC group, indoor heating control group; IT group, indoor temperature group; OT group, outdoor temperature group; SEM, standard error of means.

**Table 6 animals-12-02535-t006:** Effects of cold environments on antioxidative indexes of lambs.

Items	Groups			SEM	*p*-Value
	IHC	IT	OT		
14 d					
CAT (U/mL)	3.01 ± 0.47	2.50 ± 0.42	2.42 ± 0.41	0.43	0.119
GPx (U/mL)	186.24 ± 34.38 ^a^	166.18 ± 27.35 ^a^	93.90 ± 14.08 ^b^	20.45	0.002
T-SOD (U/mL)	14.29 ± 1.28 ^a^	12.23 ± 1.74 ^a^	6.84 ± 0.79 ^b^	1.44	0.001
T-AOC (U/mL)	0.60 ± 0.08 ^a^	0.44 ± 0.05 ^b^	0.37 ± 0.05 ^b^	0.04	0.002
MDA (nmol/mL)	1.08 ± 0.11	1.05 ± 0.15	1.18 ± 0.04	0.10	0.394
28 d					
CAT (U/mL)	2.57 ± 0.27 ^a^	1.87 ± 0.41 ^b^	1.63 ± 0.33 ^b^	0.34	0.002
GPx (U/mL)	80.16 ± 10.81 ^a^	65.20 ± 3.86 ^b^	30.40 ± 4.60 ^c^	6.81	<0.001
T-SOD (U/mL)	12.65 ± 0.81 ^a^	10.49 ± 0.49 ^b^	7.75 ± 0.66 ^c^	0.90	<0.001
T-AOC (U/mL)	0.61 ± 0.05 ^a^	0.58 ± 0.02 ^a^	0.39 ± 0.06 ^b^	0.04	<0.001
MDA (nmol/mL)	1.03 ± 0.04 ^b^	1.15 ± 0.09 ^ab^	1.25 ± 0.04 ^a^	0.05	0.017

Note: Each value is shown as the mean ± SD (*n* = 6). In the same row, values with no letter (a, b, and c) or the same letter superscripts mean no significant difference (*p* > 0.05), while those with different letter superscripts mean a significant difference (*p* < 0.05). CAT, catalase; GPx, glutathione peroxidase; T-SOD, total superoxide dismutase; T-AOC, total antioxidant capacity; MDA, malondialdehyde. IHC group, indoor heating control group; IT group, indoor temperature group; OT group, outdoor temperature group; SEM, standard error of means.

## Data Availability

The data that support the findings of this study are available from the corresponding authors upon reasonable request.

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
