# Peer review of "Influence of Cold Environments on Growth, Antioxidant Status, Immunity and Expression of Related Genes in Lambs"

_animals, 2022, doi:10.3390/ani12192535_

Round 1

Reviewer 1 Report

Dear Author(s),

In the present study, Shi et al. addressed the Influence of the Cold Environment on Growth, Antioxidant, Immunity, and Expression of Related Genes in Lambs. The manuscript presents an essential and practical contribution to improving the growth, immune, and antioxidant function of the lambs.

- In general, the aim of the study is clearly defined. The study is a survey experiment. The authors are approaching the results only by comparing the means obtained with each specific treatment or compared to previous publications.

- However, some important concerns need to arise from this study, particularly those regarding novelty and methodology.

- Line 166, please delete “Becton-Dickinson, Waltham, MA, USA” (the authors did not add anticoagulant into blood samples).

- Please provide how much µL of total RNA was used for cDNA synthesis?

- The authors performed the qRT-PCR analysis using n = ?. Please provide in Materials and Methods

-Section 2.4.4, lines 195-197: The authors mentioned that “the isolated RNA was quantitatively and qualitatively determined as described in the previous report [30]” and they used NanoDrop spectroscopy (Thermo Scientific, USA) with the ratio of absorbance at 260 and 280 nm to assess RNA quantity and quality. Actually, this is a good system to estimate nucleic acid and RNA purity and quantification. However, it does not inform about RNA integrity. How do the authors check RNA integrity? Please, explain.

- Considering that an important part of the results presented is based on qRT-PCR assays. How did the authors validate the efficiency of the reaction with the different primers? Was the specificity of the reaction checked and how?

- From Table 1 can be deduced that possibly one gene (β-actin) was used for normalization of gene expression data. Please, confirm how the results were normalized. In general, using 2 or 3 reference genes prevents the obtention of artefactual results.

- Why was standard error not taken in in this study? The SD has been taken but SD reflects variability within the sample while SE may estimate the variability across the samples of a population.

- Line 243, change ‘the’ to ‘The’.

- Figure 2: Apparently control values were set at 1 for IHC as control. The bars corresponding to all control values lack error bars. It seems like control values could correspond to single values (?). If so, controls could not be included for statistical analysis. Please, clarify.

- Section 3.3, Immune Indexes and Relevant Genes Expression. Gene names must be in italic form. Please check

- According to Figure 2, IL-2 expression displayed a significant difference between IHC and the other groups (p < 0.05) on day 28. Please confirm the statistically significant difference between IHC and ITNC treatment group.

- Figure 3: The Compare Means gene expression data are reliable when the authors compare differences in descriptive statistics across 3 factors. Why did the authors compare in single factor?

- Lines 298-307, gene names must be in italic form. Please check again in the manuscript.

Reviewer 2 Report

The paper is interesting and rather well written especially introduction and discussion but the results should be presented in more appropriate way.

Specific comments:

1.      I suggest to remove the word “non-control” in the names of ITNC and OTNC. This indicates that the experiment was not controlled.

2.      The description of ITNC group should be more detailed. How the animals were exposed to the coldness?

3.      Line 221. IHC instead HTC

4.      Additionally, please provide the average temperature during each of day (from 1 to 28)  for each of group.

5.      In each table it is not explained what the letters a and b mean. Generally all tables are difficult to interpret because is only one P-value. It would be more transparent to present the results graphically as in figures 2 and 3.

6.      Line 329. The results show different values for IHC and ITNC.

The biggest concern is that all of the measurements were not performed every day but only on days 14 and 28 of the experiment. Temperature during experiment was not constant but varied especially for OTNC group therefore it is difficult to interpret the results. The temperature varied the days and it is unknown at what time the blood was taken (between 11-16 the difference in temperatures is not great).
